# Unmet Needs Mediate the Impact of Fear of Cancer Recurrence on Screening Participation Among Cancer Survivors: A Cross-Sectional Study

**DOI:** 10.3390/healthcare13101184

**Published:** 2025-05-19

**Authors:** Mi-Lee Kim, Yeol Kim, Yu-Ri Choe

**Affiliations:** 1Department of Family Medicine, Chonnam National University Hwasun Hospital, Hwasun 58128, Republic of Korea; yuiop916@naver.com; 2National Cancer Control Institute, National Cancer Center, Goyang 10408, Republic of Korea; drheat@ncc.re.kr; 3Department of Family Medicine, Chonnam National University Medical School, Gwangju 61469, Republic of Korea

**Keywords:** cancer survivors, fear of cancer recurrence, unmet needs, mediation analysis, health screening participation

## Abstract

**Background/Objectives**: Cancer survivors require ongoing follow-up care, including regular health screening, to detect recurrence or secondary malignancies. Nonetheless, psychosocial factors may influence their participation in screening. This study aimed to investigate the associations among fear of cancer recurrence (FCR), unmet needs, and screening behavior in cancer survivors and to explore whether unmet needs mediated the relationship between FCR and health screening participation. **Methods**: Data from a cross-sectional pilot survey of 326 adult cancer survivors who completed primary cancer treatment in Korea were analyzed. Screening participation was defined as having undergone either a general health check-up or cancer screening within the past 2 years. Factors associated with screening behavior were identified using logistic regression analysis. Additionally, mediation analysis was conducted to examine the indirect effects of FCR on screening through unmet needs. **Results:** Higher income, older age, longer time since diagnosis, and fewer unmet needs were significantly associated with screening participation. FCR was not directly associated with screening but was positively associated with higher unmet needs (OR: 4.59 [95% CI: 2.66, 7.94], *p* < 0.001), which were negatively associated with screening (OR: 0.41 [95% CI: 0.20, 0.84], *p* = 0.015). The indirect effect of FCR on screening for unmet needs was statistically significant (OR: 0.25 [95% CI: 0.08, 0.85], *p* = 0.027). **Conclusions**: Unmet needs may mediate the relationship between FCR and screening behavior in cancer survivors. Addressing these needs may represent a promising strategy for improving adherence to recommended follow-up screening.

## 1. Introduction

With advances in cancer treatment and increased uptake of early detection, the cancer survival rate in Korea has steadily improved. The most recent statistics from the Korea Central Cancer Registry indicated that there were approximately 2.6 million cancer survivors in Korea as of 1 January 2023. Notably, more than 60% of these individuals survive for over 5 years following their diagnosis, reflecting a significant annual increase [1]. Consequently, the importance of post-treatment survivorship care has attracted considerable attention. Regular follow-up visits are essential for cancer survivors not only to detect recurrence or secondary malignancies early but also to manage treatment-related sequelae and promote their psychological well-being [2].

Participation in recommended screening among cancer survivors has been linked to several determinants, including health insurance coverage, household income, time since last screening, and continuity of care with a primary physician [3]. A cross-sectional study revealed the association of individual-level social risks, such as financial strain and lack of transportation, with screening non-adherence [4]. Psychological conditions such as depression and anxiety have also been reported to negatively affect screening participation in certain subgroups of cancer survivors [5]. These findings suggest that screening adherence is influenced by various factors across individual, social, and healthcare system domains. Among these factors, the fear of cancer recurrence (FCR) and unmet needs have emerged as salient psychosocial contributors [6].

FCR, defined as the ongoing worry or fear that cancer will return, can substantially influence health-related behaviors. Moderate FCR levels may serve as a motivator for regular follow-up and surveillance [7,8], whereas excessive FCR may elicit avoidant coping strategies that hinder compliance with screening recommendations [9]. Unmet needs, which encompass a lack of adequate medical, emotional, or social support, may also interfere with screening participation owing to barriers such as insufficient information, emotional distress, and financial hardship [10]. Although previous studies in Korea have independently investigated the impact of FCR and unmet needs on screening behavior [11], research on their interactive effects remains limited. Although the prevalence of FCR among cancer survivors is well established, empirical evidence clarifying the mechanisms through which FCR influences health behaviors, particularly adherence to follow-up screening, is lacking. In this context, unmet needs—modifiable yet often underrecognized psychosocial factors—may mediate the relationship between FCR and screening participation. However, few studies have conceptualized unmet needs as a mediating variable in this association. Addressing this gap is critical for informing the development of effective survivorship care strategies. Therefore, the present study aimed to investigate the independent and combined effects of FCR and unmet needs on cancer screening participation using data from the Korean Nationwide Survey for Cancer Survivorship. These findings can inform evidence-based policies and facilitate the development of targeted interventions to improve screening adherence among cancer survivors.

## 2. Materials and Methods

### 2.1. Study Design and Participants

This study used data derived from a pilot survey conducted prior to the Korean Nationwide Survey for Cancer Survivorship. The pilot survey aimed to evaluate the feasibility of the study protocol, refine the questionnaire, and validate the data collection procedures. The survey covered a wide range of topics, including treatment history, current health status, quality of life, unmet needs, health behaviors, health policy indicators, and sociodemographic characteristics. The specific domains included cancer type and stage, treatment modality and status, healthcare costs, subjective health status, mental health status, presence of comorbidities, physical activity, dietary habits, and participation in secondary cancer screening. Additionally, items assessing physical, social, and psychological needs over the past year, as well as unmet informational needs, were included.

The eligibility criteria for the study participants were as follows: adults aged ≥19 years who were diagnosed with one of the six major cancers in Korea (stomach, colorectal, liver, breast, cervical, or lung cancer) and who had survived for at least 1 year after their diagnosis. Participants were required to have completed primary cancer treatment, such as surgery, chemotherapy, or radiotherapy. Exceptions were made for individuals who were currently receiving oral chemotherapy without evidence of disease progression, who received adjuvant hormonal therapy following breast cancer treatment, or who exhibited no disease progression after receiving local treatment for liver cancer. Individuals currently undergoing treatment for recurrent or metastatic cancer and those diagnosed with terminal-stage cancer were excluded.

Recruitment was conducted in collaboration with the medical staff at four institutions in Korea—namely, a national cancer center, two major private hospitals, and a regional cancer center. A total of 377 participants were enrolled from 6 November 2023 to 29 March 2024. Data were collected by a professional survey agency via either in-person interviews or self-administered online questionnaires, based on participant preferences. Out of 377 recruited individuals, 339 completed the survey. After excluding nine participants with stage IV cancer and four individuals diagnosed with non-major cancers (e.g., thyroid cancer, carcinoid sarcoma), 326 participants were finally included in the analysis (Figure 1).

The study protocol was approved by the Institutional Review Board of the Chonnam National University Hwasun Hospital (IRB no. CNUHH-2023-20). Informed consent was obtained through face-to-face interviews with trained research nurses using a web-based consent form.

### 2.2. Variables

This study aimed to identify factors associated with health screening participation, defined as the receipt of either a general health check-up or cancer screening within the past 2 years, in cancer survivors.

Sociodemographic variables included age, sex, marital status, educational level, monthly household income, comorbidities, and employment status. Marital status was dichotomized as “single (divorced, or widowed)” and “married and cohabiting”. Educational level was categorized into middle school graduate or lower, high school graduate, and college graduate or higher. Monthly household income was divided into quartiles (1Q–4Q) based on the distribution of all respondents. Comorbidity status was defined based on the presence of major chronic diseases such as hypertension, diabetes, and cardiovascular disease and was categorized into “none” and “one or more”. Employment status was categorized based on current economic activity into “employed” and “unemployed”. Cancer-related clinical variables included cancer type, stage, treatment history (surgery, chemotherapy, radiotherapy, hormone therapy, and procedures), and time since diagnosis. Cancer stage was classified from stage 0 to stage III, and cases in which the respondent reported not knowing the stage were categorized as “unknown”. Each treatment modality was treated as an independent variable, and multiple responses were allowed. Time since diagnosis was categorized into “1.0–4.5 years” and “4.5–10 years”.

FCR was assessed using a single-item question: “Are you worried that your cancer might come back?” Responses were collected on a four-point Likert scale (“not at all”, “slightly”, “quite”, and “very”) [12]. For analysis, responses were dichotomized into two categories: low fear (“not at all” and “slightly”) and high fear (“quite” and “very”). Unmet needs were evaluated using items selected from the Cancer Patient Needs Assessment Tool, originally developed by the Ministry of Health and Welfare [13]. Items from the following four domains were employed: information and education (10 items), psychological needs (10 items), symptom management (12 items), and social support (two items). Each item was rated on a 4-point scale from 1 to 4 (“not needed”, “somewhat needed”, “moderately needed”, and “greatly needed”), and scores were summed to calculate the total unmet needs score. Subsequently, the respondents were categorized into quartiles based on their total score, and those in the highest quartile (score ≥ 110.0) were deemed to have high unmet needs. Health screening participation was assessed using two binary items: “Had you undergone a general health check-up in the past 2 years?” and “Had you undergone a cancer screening in the past 2 years?” Participants who responded “yes” to either item were classified as having participated in the screening.

### 2.3. Statistical Analysis

Baseline characteristics of the study population were summarized using descriptive statistics. Continuous and categorical variables were expressed as means with standard deviations and as frequencies with percentages, respectively. Sociodemographic, clinical, and psychological characteristics were examined.

Group differences according to health screening status were evaluated using an independent *t*-test for continuous variables and a chi-square test for categorical variables. Multivariate logistic regression analysis was conducted to identify factors independently associated with health screening participation. Variables were selected based on their clinical relevance and statistical significance in univariate analyses.

Mediation analysis was also conducted to determine whether total unmet needs mediated the association between FCR and screening participation. This analysis followed the approach proposed by Baron and Kenny, which required the establishment of (i) a significant relationship between the independent variable (i.e., FCR) and dependent variable (i.e., screening); (ii) a significant relationship between the independent variable and mediator (i.e., unmet needs); and (iii) a significant association between the mediator and dependent variable after adjustment for the independent variable. Statistical significance of the indirect effect was evaluated using the Sobel test.

All statistical analyses were performed using Python version 3.10 (Python Software Foundation, Wilmington, NC, United States) [14] with relevant libraries, including pandas [15], statsmodels [16], and SciPy [17]. Statistical significance was set at *p* < 0.05.

## 3. Results

### 3.1. Sociodemographic and Clinical Characteristics of the Participants

This study included 326 cancer survivors with a mean age of 57.5 ± 10.7 years (Table 1). Female and male participants accounted for 69.9% (*n* = 228) and 30.1% (*n* = 98), respectively. Most participants were married or living with a partner (*n* = 240, 73.6%), whereas 86 (26.4%) were single, divorced, or widowed. With respect to educational attainment, 34 participants (10.4%) had completed middle school or lower, 112 (34.4%) had graduated from high school, and 180 (55.2%) had attained a college degree or higher. Monthly household income was categorized into quartiles, with each quartile representing approximately one-quarter of the study population.

Overall, 98 participants (30.1%) reported having at least one chronic condition (hypertension, diabetes, or cardiovascular disease). Furthermore, 183 participants (56.1%) were employed. Regarding clinical characteristics, the most common cancer type was breast cancer (*n* = 105, 32.2%), followed by stomach cancer (23.0%), colorectal cancer (15.0%), lung cancer (12.6%), cervical or uterine cancer (9.5%), and liver cancer (7.7%). The most frequently reported stage at diagnosis was stage I (40.5%), followed by stage II (23.9%), stage III (22.4%), and stage 0 (7.4%); 19 (5.8%) participants were unaware of their cancer stage.

The treatment history was collected as a multiple-response item. Surgical treatment was the most common (*n* = 308, 94.5%), followed by chemotherapy (49.4%), radiotherapy (39.6%), hormone therapy (18.4%), and procedure-based treatment (2.5%). Time since cancer diagnosis was 1.0 to <4.5 years in 53.4% of the participants and 4.5–10 years in 46.6%.

FCR was assessed using a single-item measure. Among the participants, 27 (8.3%) reported no fear, whereas 147 (45.1%), 78 (23.9%), and 74 (22.7%) expressed slight, moderate, and severe fear, respectively. Based on the total unmet needs score, 83 (25.5%) participants were classified as having high unmet needs (top quartile).

The distribution of unmet needs across the four assessed domains (namely, information and education, psychological needs, symptom management, and social support) is shown in Appendix A. Out of the 326 cancer survivors, 285 (87.4%) reported having undergone either a general health check-up or cancer screening within the past 2 years, whereas 41 (12.6%) reported not undergoing screening during this period.

### 3.2. Group Comparison by Health Screening

A comparison of participant characteristics according to health-screening status revealed several significant differences (Table 2). Screening participation was significantly associated with lower monthly household income (*p* = 0.008), cancer type (*p* = 0.031), cancer stage (*p* = 0.010), and history of chemotherapy (*p* = 0.037). Furthermore, longer time since diagnosis (4.5 ≤ years ≤ 10) demonstrated higher screening participation. Among psychosocial variables, higher unmet needs were associated with lower screening rates (*p* = 0.020), whereas FCR was not significantly associated with screening participation (*p* = 0.817).

### 3.3. Multivariate Analysis of Health Screening Participation

Multivariate logistic regression analysis identified several factors independently associated with screening participation (Table 3). Compared with the lowest income group (1Q), participants in the second (odds ratio [OR]: 5.17 [95% confidence interval (CI): 1.72, 15.53], *p* = 0.003), third (OR: 3.01 [95% CI: 1.20, 7.54], *p* = 0.019), and fourth (OR: 3.75 [95% CI: 1.28, 10.98], *p* = 0.016) income quartiles were significantly more likely to undergo screening. Age was positively associated with screening participation (OR: 1.02 [95% CI: 0.98, 1.06], *p* = 0.021). Overall, cancer stage was significantly associated with screening (*p* = 0.006); however, none of the individual stage comparisons reached statistical significance. High unmet needs showed a negative association with screening participation (OR: 0.43 [95% CI: 0.19, 0.97], *p* = 0.042).

### 3.4. Mediation Effect of Unmet Needs Between FCR and Health Screening

Mediation analysis revealed that unmet needs significantly mediated the association between FCR and screening participation (Figure 2). Specifically, higher FCR levels were associated with increased unmet needs (OR: 4.59 [95% CI: 2.66, 7.94], *p* < 0.001), and higher unmet needs were negatively associated with screening participation (OR: 0.41 [95% CI: 0.20, 0.84], *p* = 0.015). The indirect effect of FCR on screening through unmet needs was statistically significant (OR: 0.25 [95% CI: 0.08, 0.85], *p* = 0.027), whereas the direct effect of FCR on screening was not significant (OR: 1.14 [95% CI: 0.56, 2.31], *p* = 0.724). These findings suggest that the influence of FCR on screening behavior may operate indirectly via unmet needs, underscoring the importance of addressing unmet supportive care needs in cancer survivors.

## 4. Discussion

This study explored the factors influencing health-screening participation in cancer survivors, with a focus on the mediating role of unmet needs in the relationship between FCR and screening behavior. The findings indicate that unmet needs significantly mediate this relationship, suggesting a potential pathway through which addressing the psychosocial and healthcare needs of cancer survivors may enhance their adherence to recommended follow-up care.

In this study, health screening participation was defined as having undergone either a general health check-up or cancer screening within the past 2 years. In Korea, health screening programs are structured around the life course and are divided into two major components for adults: general health check-ups and national cancer screening [18,19]. Although the general health check-up focuses on the early detection and prevention of chronic conditions such as hypertension and diabetes, the cancer screening program targets six major cancers (gastric, colorectal, liver, breast, cervical, and lung cancers) based on age, sex, and individual risk factors. Although these two programs differ in purpose, disease targets, and implementation, in practice, they are often received concurrently, with little clear distinction made by patients. In our dataset, 99.0% of the participants who underwent cancer screening also reported having undergone a general health check-up, indicating a high degree of overlap in screening behavior. Additionally, 67.1% of those who underwent a general health check-up participated in cancer screening, suggesting that a significant proportion of individuals engage in both types of screening concurrently. These findings imply that, in real-world settings, individuals may not clearly differentiate between the two programs in their health-seeking behavior. Furthermore, prior studies suggest that public awareness and understanding of cancer screening remain limited, and some individuals may confuse the purpose, scope, or target diseases of screening programs. This may be partially explained by the structural overlap between the two programs. For instance, upper endoscopy and breast ultrasound, although primarily considered cancer screening modalities, are sometimes included in general health check-ups [20]. Therefore, to more accurately capture actual screening participation, we defined “health screening” as having undergone either a general health check-up or a cancer screening within the past 2 years. It is important to acknowledge, however, that the underlying motivations for participating in general health check-ups versus cancer-specific screenings may differ. Although this integrated definition captures the structural and behavioral overlap between the two programs, it may have constrained our ability to identify distinct psychological or behavioral determinants associated with each modality. Future studies should consider disaggregating general and cancer screening behaviors to more precisely elucidate their potentially divergent motivational pathways.

Higher income, older age, and longer time since diagnosis were positively associated with screening participation. In particular, lower monthly household income was associated with lower screening participation, which is consistent with the socioeconomic disparities observed in national cancer screening programs for major cancers, such as breast and colorectal cancers, in Korea [21,22,23]. The significance of this study lies in its confirmation that similar patterns persist within the cancer survivor population in Korea. These findings confirm that similar disparities persist among cancer survivors.

Although FCR was not directly associated with screening participation, it was indirectly linked through unmet needs, which were significantly associated with reduced screening rates. These findings suggest that FCR may influence health behaviors not independently, but by shaping survivors’ perceived medical and psychosocial support needs. This result is consistent with those of prior studies, indicating that unaddressed emotional and practical needs may lead to decreased utilization of health services [24,25]. The significant association observed between FCR and unmet needs further suggests that FCR is not merely a transient emotional state, but rather reflects specific and tangible unmet needs experienced by survivors [6,26,27]. This finding aligns with previous research demonstrating that elevated FCR among breast cancer survivors may result in healthcare avoidance and delays in medical decision-making [28]. The present study adds to this literature by statistically validating the mediating role of unmet needs through a structured mediation model. Conversely, some studies have reported that higher FCR may lead to increased healthcare utilization [8]. These discrepancies may be explained by extrinsic factors such as variations in healthcare systems, insurance structures, cultural norms, and cancer-specific follow-up recommendations across countries. Notably, this study was conducted within the context of Korea’s universal healthcare system, and such environmental characteristics may have contributed to the absence of a direct association between FCR and screening participation. Taken together, this study demonstrates that FCR may influence screening behaviors among cancer survivors not directly, but indirectly through unmet needs. These findings highlight the importance of addressing FCR along with the associated medical and psychosocial needs it may trigger, rather than focusing solely on FCR as an isolated emotional response. By validating the mediating role of unmet needs, this study contributes to a clearer understanding of the psychosocial mechanisms underlying screening behavior.

The findings of this study offer important implications for clinical practice and survivorship care. Identifying and addressing unmet needs—particularly those related to psychological distress and informational deficits—may enhance adherence to recommended screenings among survivors experiencing FCR. Incorporating routine assessments of unmet needs into survivorship care may help clinicians identify individuals at greater risk of disengagement from follow-up care. Furthermore, comprehensive, multidisciplinary interventions that target emotional and practical support needs could mitigate the psychological burden affecting health behaviors. These insights support a patient-centered, need-responsive model of follow-up care that may be applicable across diverse cancer survivor populations.

Although carcinoma in situ (stage 0) is associated with an excellent prognosis, individuals with this diagnosis were included in the present study given the behavioral focus of our investigation. As formally recognized cancer survivors, patients with stage 0 disease may still experience substantial psychological distress—such as FCR—and encounter various unmet informational or emotional needs. Their inclusion reflects the full psychosocial spectrum of survivorship and supports a more comprehensive understanding of the factors that influence engagement in follow-up care across all cancer stages. Thus, their inclusion is justified in psychosocial and behavioral cancer research, as it ensures a more complete representation of the survivorship experience across the full spectrum of disease stages.

Despite these strengths, the present study has some limitations. First, because of its cross-sectional design, causal relationships among the key variables could not be established. Second, key variables such as screening participation, unmet needs, and FCR were measured using self-reported questionnaires, which might have introduced recall and reporting biases. In particular, the use of self-reported measures may also have introduced social desirability effects, potentially influencing how participants responded to questions about psychological distress and screening behavior. This could have affected the observed associations and should be considered when interpreting the findings. Third, as a pilot study, the sample size was limited, and future research should involve larger and more representative populations to enhance generalizability. Fourth, although this study defined high unmet needs using a quartile-based threshold, subsequent studies may benefit from an analysis of unmet needs as a continuous variable or by domain to more precisely capture their quantitative impact. Building on this, it is also important to consider whether distinct domains of unmet needs differentially mediate the relationship between FCR and screening behavior. Although this study focused on the total unmet needs score to examine the mediating pathway, unmet needs were assessed across four domains: informational and educational, psychological, symptom-related, and social support. Future research should explore domain-specific mediation models to identify which types of unmet needs most strongly influence screening behavior. Such analyses could guide the development of tailored interventions that address the most impactful areas of support in cancer survivorship care. Finally, because this study was conducted within the context of Korea’s universal healthcare system and cultural environment, caution should be exercised when generalizing the findings to other countries and healthcare systems. To address these limitations, future research should adopt longitudinal designs to trace the causal pathways among FCR, unmet needs, and screening behaviors over time. Additionally, intervention-based studies should be conducted to investigate whether psychosocial interventions targeting FCR or unmet needs can effectively promote positive changes in health behaviors. Comparative studies across diverse cultural and healthcare contexts should also be performed to strengthen the external validity and cross-national applicability of these findings.

## 5. Conclusions

This study suggests that FCR may indirectly influence the screening behavior in cancer survivors through unmet supportive care needs. Although FCR alone was not directly associated with screening participation, it was significantly linked to increased unmet needs, which, in turn, were associated with lower screening uptake. These findings highlight the importance of addressing not only emotional distress but also the broader medical and psychosocial needs that emerge during cancer survivorship. Notably, the results imply that interventions targeting unmet needs could serve as a practical and potentially effective strategy to enhance engagement in recommended screenings. This warrants further investigation through longitudinal research. Rather than focusing solely on regulating emotional responses, survivorship care strategies should incorporate systems that identify, assess, and manage supportive care needs. At the policy level, routine screening for unmet needs in clinical practice, coupled with timely psychosocial and navigational support, may help reduce disparities and improve the long-term health outcomes of cancer survivors.

## Figures and Tables

**Figure 1 healthcare-13-01184-f001:**
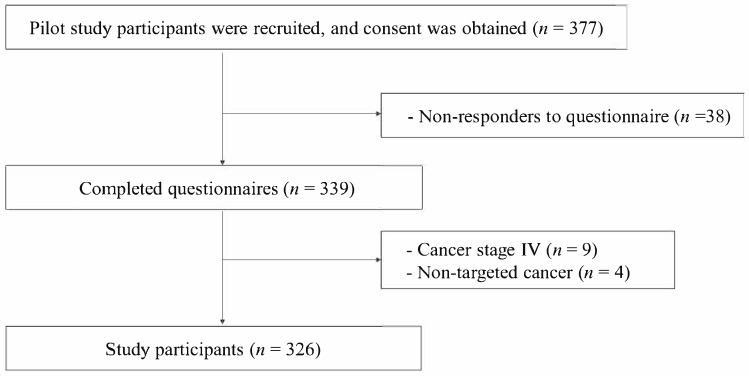
Flowchart of the participants.

**Figure 2 healthcare-13-01184-f002:**
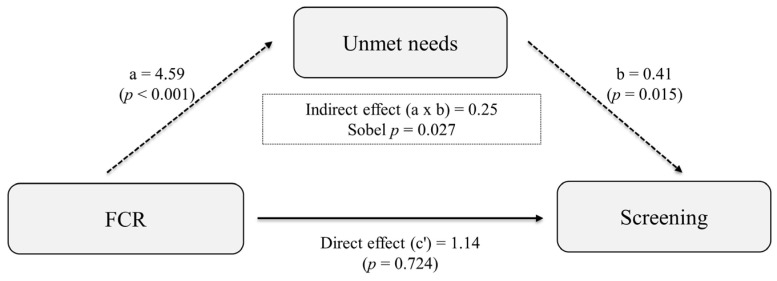
Mediation analysis of unmet needs in the relationship between FCR and health screening. FCR was positively associated with unmet needs (*a* = 4.59 [95% CI: 2.66, 7.94], *p* < 0.001), and unmet needs were negatively associated with screening participation (*b* = 0.41 [95% CI: 0.20, 0.84], *p* = 0.015). The indirect effect of FCR on screening (*a* × *b* = 0.25 [95% CI: 0.08, 0.85]) was statistically significant (Sobel *p* = 0.027), whereas the direct effect of FCR on screening was not (*c*′ = 1.14 [95% CI: 0.56, 2.31], *p* = 0.724). FCR, fear of cancer recurrence.

**Table 1 healthcare-13-01184-t001:** Sociodemographic, clinical, and psychological characteristics of the participants.

Variables	*n* = 326
Age	57.5 ± 10.7
Sex	
Male	98 (30.1)
Female	228 (69.9)
Marital status	
Single	86 (26.4)
Married and cohabiting	240 (73.6)
Educational level	
Middle school graduate or lower	34 (10.4)
High school graduate	112 (34.4)
College graduate or higher	180 (55.2)
Monthly household income ^a^	
1Q	84 (25.8)
2Q	80 (24.5)
3Q	91 (27.9)
4Q	71 (21.8)
Comorbidity	
0	228 (69.9)
≥1	98 (30.1)
Employment status	
Unemployed	143 (43.9)
Employed	183 (56.1)
Cancer type	
Breast	105 (32.2)
Stomach	75 (23.0)
Colorectal	49 (15.0)
Lung	41 (12.6)
Cervical or uterine	31 (9.5)
Liver	25 (7.7)
Cancer stage	
0	24 (7.4)
I	132 (40.5)
II	78 (23.9)
III	73 (22.4)
Unknown	19 (5.8)
Surgery (yes)	308 (94.5)
Chemotherapy (yes)	161 (49.4)
Radiotherapy (yes)	129 (39.6)
Hormone therapy (yes)	60 (18.4)
Procedure (yes)	8 (2.5)
Time since diagnosis	
1.0 ≤ years < 4.5	174 (53.4)
4.5 ≤ years ≤ 10	152 (46.6)
FCR	
Not at all	27 (8.3)
Slightly	147 (45.1)
Quite	78 (23.9)
Very	74 (22.7)
Unmet needs ^b^	
1–3Q	243 (74.5)
4Q (high unmet needs)	83 (25.5)
Health screening (yes)	285 (87.4)

Data are expressed as mean ± standard deviation or number (%). FCR, fear of cancer recurrence. ^a^ Monthly household income was divided into four groups according to the quartiles of the total respondent distribution. ^b^ Total scores across the four domains were summed and categorized into quartiles.

**Table 2 healthcare-13-01184-t002:** Comparison of sociodemographic, clinical, and psychological characteristics according to screening status.

	Screening (No) *n* = 41	Screening (Yes) *n* = 285	*p*-Value
Age	55.8 ± 11.9	57.8 ± 10.5	0.329
Sex			0.506
Male	10 (10.2%)	88 (89.8%)	
Female	31 (13.6%)	197 (86.4%)	
Marital status			0.163
Single	15 (17.4%)	71 (82.6%)	
Married and cohabiting	26 (10.8%)	214 (89.2%)	
Educational level			0.415
Middle school graduate or lower	6 (17.6%)	28 (82.4%)	
High school graduate	16 (14.3%)	96 (85.7%)	
College graduate or higher	19 (10.6%)	161 (89.4%)	
Monthly income ^a^			0.008
1Q	19 (22.6%)	65 (77.4%)	
2Q	5 (6.2%)	75 (93.8%)	
3Q	11 (12.1%)	80 (87.9%)	
4Q	6 (8.5%)	65 (91.5%)	
Comorbidity			0.949
0	28 (12.3%)	200 (87.7%)	
≥1	13 (13.3%)	85 (86.7%)	
Employment status			0.610
Unemployed	20 (14.0%)	123 (86.0%)	
Employed	21 (11.5%)	162 (88.5%)	
Cancer type			0.031
Breast	8 (7.6%)	97 (92.4%)	
Stomach	10 (13.3%)	65 (86.7%)	
Colorectal	10 (20.4%)	39 (79.6%)	
Lung	2 (4.9%)	39 (95.1%)	
Cervical or uterine	8 (25.8%)	23 (74.2%)	
Liver	3 (12.0%)	22 (88.0%)	
Cancer stage			0.010
0	8 (7.6%)	97 (92.4%)	
I	10 (20.4%)	39 (79.6%)	
II	10 (13.3%)	65 (86.7%)	
III	2 (4.9%)	39 (95.1%)	
Unknown	3 (12.0%)	22 (88.0%)	
Surgery			1.000
No	2 (11.1%)	16 (88.9%)	
Yes	39 (12.7%)	269 (87.3%)	
Chemotherapy			0.037
No	14 (8.5%)	151 (91.5%)	
Yes	27 (16.8%)	134 (83.2%)	
Radiotherapy			0.663
No	23 (11.7%)	174 (88.3%)	
Yes	18 (14.0%)	111 (86.0%)	
Hormone therapy			0.189
No	37 (13.9%)	229 (86.1%)	
Yes	4 (6.7%)	56 (93.3%)	
Procedure			0.585
No	41 (12.9%)	277 (87.1%)	
Yes	0 (0.0%)	8 (100.0%)	
Time since diagnosis			0.027
1.0 ≤ years < 4.5	29 (16.7%)	145 (83.3%)	
4.5 ≤ years ≤ 10	12 (7.9%)	140 (92.1%)	
FCR			0.817
Not at all	2 (7.4%)	25 (92.6%)	
Slightly	20 (13.6%)	127 (86.4%)	
Quite	9 (11.5%)	69 (88.5%)	
Very	10 (13.5%)	64 (86.5%)	
Unmet needs ^b^			0.020
1–3Q	24 (9.9%)	219 (90.1%)	
4Q (high unmet needs)	17 (20.5%)	66 (79.5%)	

Data are expressed as mean ± standard deviation or number (%). FCR, fear of cancer recurrence. ^a^ Monthly household income was divided into four groups according to the quartiles of the total respondent distribution. ^b^ Total scores across the four domains were summed and categorized into quartiles.

**Table 3 healthcare-13-01184-t003:** Factors associated with screening in the multivariate logistic regression analysis.

	OR (95% CI)	*p*-Value
Age	1.02 (0.98, 1.06)	0.021
Sex	0.94 (0.39, 2.27)	0.894
Monthly household income		0.006
1Q	Reference	
2Q	5.17 (1.72, 15.53)	0.003
3Q	3.01 (1.20, 7.54)	0.019
4Q	3.75 (1.28, 10.98)	0.016
Cancer stage		0.006
0	Reference	
I	1.34 (0.26, 7.01)	0.731
II	0.76 (0.14, 4.09)	0.747
III	0.25 (0.05, 1.29)	0.098
Unknown	0.80 (0.09, 7.38)	0.847
Time since diagnosis (4.5 ≤ years ≤ 10)	2.49 (1.14, 5.44)	0.327
FCR (quite, very)	1.25 (0.57, 2.76)	0.582
High unmet needs	0.43 (0.19, 0.97)	0.042

FCR, fear of cancer recurrence.

## Data Availability

The data presented in this study are available upon reasonable request from the corresponding author. The data are not publicly available because of ethical restrictions and the potential for re-identification of participants.

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
