# Peer review of "Unmet Needs Mediate the Impact of Fear of Cancer Recurrence on Screening Participation Among Cancer Survivors: A Cross-Sectional Study"

_healthcare, 2025, doi:10.3390/healthcare13101184_

Round 1
Reviewer 1 Report
Comments and Suggestions for Authors
This study examines how unmet needs mediate the effect of fear of cancer recurrence on screening participation among Korean cancer survivors, highlighting unmet needs as key targets to improve follow-up care adherence.
- The definition of health screening as either a general health check-up or cancer-specific test may dilute the specificity of the outcome variable. These should ideally be analyzed separately, or the overlap more critically examined, as the motivations for attending each may differ.
- While acknowledged in the discussion, the cross-sectional nature of the data prevents firm conclusions about mediation. The manuscript would benefit from a more cautious tone when interpreting the directionality of effects, especially in the abstract and conclusion.
- The unmet needs score was dichotomized into quartiles, yet no domain-specific analysis was presented. Exploring which specific types of unmet needs (e.g., psychological vs. informational) mediate the relationship most strongly would yield more actionable insights.
- Both FCR and screening adherence were self-reported, introducing recall and social desirability biases. This limitation should be emphasized more clearly in both the Discussion and Limitations sections.
- Several sections (especially the Discussion) are repetitive and would benefit from stylistic tightening. For instance, the distinction between the direct and indirect effects of FCR is explained multiple times in similar language. In the discussion please cite: doi: 10.3390/cancers16223766.
Author Response
- The definition of health screening as either a general health check-up or cancer-specific test may dilute the specificity of the outcome variable. These should ideally be analyzed separately, or the overlap more critically examined, as the motivations for attending each may differ.
Response) We thank the reviewer for this important and insightful comment. We fully agree that general health check-ups and cancer screenings differ in their intended purposes and that the motivations for attending each may not be identical. In response, we have revised the Discussion section to explicitly acknowledge this limitation and to clarify the rationale for our integrated definition of health screening.
Specifically, we added a detailed explanation of the Korean national health screening system, in which general health check-ups and cancer screenings are frequently administered concurrently, resulting in substantial structural and behavioral overlap. In our dataset, 99.0% of participants who underwent cancer screening also received a general health check-up, and 67.1% of those who had a general health check-up also received cancer screening. These findings support our decision to adopt an integrated definition, as it more accurately reflects real-world screening behavior in the Korean context.
Nevertheless, we now explicitly acknowledge in the revised Discussion that this integrated approach may limit the ability to distinguish the unique psychological or behavioral determinants associated with each screening type.
The following sentences have been added to the Discussion section:
- (page 10, lines 274-278):
“Additionally, 67.1% of those who underwent a general health check-up participated in cancer screening, suggesting that a significant proportion of individuals engage in both types of screening concurrently. These findings imply that, in real-world settings, individuals may not clearly differentiate between the two programs in their health-seeking behavior.”
- (page 10, lines 286-292):
"It is important to acknowledge, however, that the underlying motivations for participating in general health check-ups versus cancer-specific screenings may differ. Although this integrated definition captures the structural and behavioral overlap between the two programs, it may have constrained our ability to identify distinct psychological or behavioral determinants associated with each modality. Future studies should consider disaggregating general and cancer screening behaviors to more precisely elucidate their potentially divergent motivational pathways."
We hope this clarification addresses the reviewer’s concern.
- While acknowledged in the discussion, the cross-sectional nature of the data prevents firm conclusions about mediation. The manuscript would benefit from a more cautious tone when interpreting the directionality of effects, especially in the abstract and conclusion.
Response) Thank you for your important comment. We agree that the cross-sectional nature of the data limits causal inference, particularly regarding mediation pathways. In response, we have revised the manuscript to adopt a more cautious tone when discussing the directionality of the observed associations. We now emphasize that the findings suggest, rather than confirm, potential mediation and encourage longitudinal research to further investigate causal mechanisms.
The updated text is as follows:
- (Abstract, page 1, lines 27-29)
“Unmet needs may mediate the relationship between FCR and screening behavior in cancer survivors. Addressing these needs may represent a promising strategy for improving adherence to recommended follow-up screening.”
- (Discussion, page 9, lines 259-262)
“The findings indicate that unmet needs significantly mediate this relationship, suggesting a potential pathway through which addressing the psychosocial and healthcare needs of cancer survivors may enhance their adherence to recommended follow-up care.”
- (Conclusion, page 12, lines 376-384)
“This study suggests that FCR may indirectly influence the screening behavior in cancer survivors through unmet supportive care needs. Although FCR alone was not directly associated with screening participation, it was significantly linked to increased unmet needs, which, in turn, were associated with lower screening uptake. These findings highlight the importance of addressing not only emotional distress but also the broader medical and psychosocial needs that emerge during cancer survivorship. Notably, the results imply that interventions targeting unmet needs could serve as a practical and potentially effective strategy to enhance engagement in recommended screenings. This warrants further investigation through longitudinal research.”
We are grateful for the reviewer’s suggestion, which we believe has meaningfully contributed to improving the clarity and quality of our work.
- The unmet needs score was dichotomized into quartiles, yet no domain-specific analysis was presented. Exploring which specific types of unmet needs (e.g., psychological vs. informational) mediate the relationship most strongly would yield more actionable insights.
Response) We appreciate the reviewer’s thoughtful suggestion. We fully agree that examining domain-specific unmet needs could provide more targeted insights into how distinct types of needs contribute to the relationship between FCR and screening behavior. In our study, unmet needs were assessed across four validated domains: informational and educational needs, psychological needs, symptom-related needs, and social support needs.
To preserve statistical power and parsimony in our mediation model, we used the total unmet needs score in the main analysis. However, we recognize the value of exploring domain-specific mediation pathways. Accordingly, we have added a statement to the Discussion section (page 11, lines 357-365) to acknowledge this limitation and to suggest domain-specific mediation analyses as an important avenue for future research.
“Building on this, it is also important to consider whether distinct domains of unmet needs differentially mediate the relationship between FCR and screening behavior. Although this study focused on the total unmet needs score to examine the mediating pathway, unmet needs were assessed across four domains: informational and educational, psychological, symptom-related, and social support. Future research should explore domain-specific mediation models to identify which types of unmet needs most strongly influence screening behavior. Such analyses could guide the development of tailored interventions that address the most impactful areas of support in cancer survivorship care.”
Although we chose to retain our original analytical design based on the objectives and framework of the study, we deeply appreciate the reviewer’s thoughtful recommendation and hope that our rationale is understandable.
- Both FCR and screening adherence were self-reported, introducing recall and social desirability biases. This limitation should be emphasized more clearly in both the Discussionand Limitationssections.
Response) We thank the reviewer for this thoughtful comment. We fully acknowledge that recall and social desirability biases are important considerations when interpreting findings derived from self-reported measures. Although these limitations were mentioned in the original manuscript, we have revised and expanded the Discussion section to more explicitly address how such biases may have influenced participants’ responses—particularly with regard to the assessment of FCR and screening participation. In response to the reviewer’s suggestion, we have added the following sentences to the Discussion section (page 11, lines 349-352):
“In particular, the use of self-reported measures may also have introduced social desirability effects, potentially influencing how participants responded to questions about psychological distress and screening behavior. This could have affected the observed associations and should be considered when interpreting the findings.”
We hope this clarification satisfactorily addresses the reviewer’s concern.
- Several sections (especially the Discussion) are repetitive and would benefit from stylistic tightening. For instance, the distinction between the direct and indirect effects of FCR is explained multiple times in similar language. In the discussion please cite: doi: 10.3390/cancers16223766.
Response) We sincerely thank the reviewer for this valuable comment. In response, we have revised the Discussion section to eliminate repetitive phrasing and enhance stylistic clarity, particularly regarding the distinction between the direct and indirect effects of FCR.
Specifically, we replaced previously detailed and overlapping explanations of FCR’s indirect impact on screening participation via unmet needs with more concise and structurally streamlined versions. The revised text now clearly conveys the key message while avoiding redundancy. Similarly, the concluding paragraph of the Discussion has been rewritten to emphasize the central mediation finding in a more succinct and refined manner. We believe these revisions align with the reviewer’s suggestions and improve the manuscript’s clarity and readability.
The updated text is as follows (pages 10-11, lines 318-324):
“Taken together, this study demonstrates that FCR may influence screening behaviors among cancer survivors not directly, but indirectly through unmet needs. These findings highlight the importance of addressing FCR along with the associated medical and psychosocial needs it may trigger, rather than focusing solely on FCR as an isolated emotional response. By validating the mediating role of unmet needs, this study contributes to a clearer understanding of the psychosocial mechanisms underlying screening behavior.”
Additionally, the corresponding explanatory paragraph has been revised as (page 10, lines 300-304):
“Although FCR was not directly associated with screening participation, it was indirectly linked through unmet needs, which were significantly associated with reduced screening rates. These findings suggest that FCR may influence health behaviors not independently, but by shaping survivors' perceived medical and psychosocial support needs.”
Finally, we sincerely thank the reviewer for suggesting this reference. After carefully reviewing the cited article (doi: 10.3390/cancers16223766), we found that its primary focus lies in exploring genomic mechanisms, specifically the role of Y chromosome loss in the development and progression of urologic cancers. Although the study provides important insights into cancer biology, its scope is not directly aligned with the psychosocial and behavioral pathways explored in our research—namely, the indirect effect of fear of cancer recurrence (FCR) on screening behavior via unmet needs. Given this thematic difference, we respectfully chose not to include this reference. Instead, we ensured that our Discussion is supported by literature that more directly relates to the conceptual framework and mediation model employed in our study.
Reviewer 2 Report
Comments and Suggestions for Authors
Authors present a work addressing: ‘Unmet needs mediate the impact of fear of cancer recurrence on screening participation among cancer survivors: A cross-sectional study’.The aim of the study was to analyze the associations among fear of cancer recurrence (FCR), unmet needs, and screening behavior in cancer survivors and to explore whether unmet needs mediated the relationship between FCR and health screening participation. The general conclusion demonstrates that unmet needs mediate the negative impact of FCR on screening behavior in cancer survivors. Addressing unmet needs may improve adherence to recommended follow-up screening. I have following minor comments:
General: This is an interesting and well-structured article. The authors present their arguments clearly and support them with relevant evidence, making the findings easy to follow and engaging to read.
Major
1. The abstract is clearly written and includes the most relevant information, providing a concise overview of the study's aims, methods, and main results.
2. The introduction is well-written and informative. Although fear of cancer recurrence is a well-documented concern among cancer survivors.Thus, to further strengthen of introduction, the addition of a brief paragraph explaining the unique nature of the study and the authors’ motivation for exploring this topic would be beneficial. This could include 2–3 sentences highlighting the research gap addressed or the practical relevance of the issue.
3. It may be advisable to exclude patients with clinical stage 0 cancer, as this stage represents carcinoma in situ and is associated with a nearly 100% survival rate. Including such cases could potentially skew the findings of the status-based analysis. Unless the authors have a strong justification for including these patients in the study group, they should provide a clear rationale for their inclusion.
4. According to the clinical classification of tumors breast, lung, colorectal, etc. (as updated by the American Joint Committee on Cancer (AJCC) in 2018 for breast cancer staging), stages are represented by Roman numerals (I-IV), while the degree of histological malignancy (grade) is expressed using Arabic numerals (1-3). check the link: http://www.breastsurgeonsweb.com/wp-content/uploads/downloads/2020/10/AJCC-Breast-Cancer-Staging-System.pdf.
5. In the study group, the largest population consisted of women with breast cancer. I suggest adding information on the molecular (biological) subtype of breast cancer to Table. 6. Please add a paragraph in the discussion section explaining how the results can be applied in practice. Additionally, clarify why your work is valuable in the field and whether any general recommendations can be made based on your research.
Author Response
- The abstract is clearly written and includes the most relevant information, providing a concise overview of the study's aims, methods, and main results.
Response) Thank you for your positive feedback. We have carefully reviewed the section once more to ensure clarity and accuracy, in alignment with the overall revisions to the manuscript.
The introduction is well-written and informative. Although fear of cancer recurrence is a well-documented concern among cancer survivors. Thus, to further strengthen of introduction, the addition of a brief paragraph explaining the unique nature of the study and the authors’ motivation for exploring this topic would be beneficial. This could include 2–3 sentences highlighting the research gap addressed or the practical relevance of the issue.
Response) Thank you for your thoughtful and constructive comment. In response, we have added a brief paragraph at the end of the Introduction to clarify the unique contribution of our study and to highlight the motivation for exploring the relationship between fear of cancer recurrence (FCR), unmet needs, and screening adherence. Specifically, we emphasized the scarcity of empirical studies examining the pathways linking FCR to health behaviors and the potential mediating role of unmet needs. This addition aims to more clearly articulate the research gap and highlight the practical relevance of our findings within the context of survivorship care. The newly added paragraph in the Introduction section is as follows (page 2, lines 61-68):
“Although the prevalence of FCR among cancer survivors is well established, empirical evidence clarifying the mechanisms through which FCR influences health behaviors, particularly adherence to follow-up screening, is lacking. In this context, unmet needs—modifiable yet often underrecognized psychosocial factors—may mediate the relationship between FCR and screening participation. However, few studies have conceptualized unmet needs as a mediating variable in this association. Addressing this gap is critical for informing the development of effective survivorship care strategies.”
We hope this addition adequately addresses your suggestion and enhances the clarity and impact of the Introduction.
- It may be advisable to exclude patients with clinical stage 0 cancer, as this stage represents carcinoma in situ and is associated with a nearly 100% survival rate. Including such cases could potentially skew the findings of the status-based analysis. Unless the authors have a strong justification for including these patients in the study group, they should provide a clear rationale for their inclusion.
Response) We appreciate the reviewer’s thoughtful comment regarding the inclusion of patients with clinical stage 0 cancer. We agree that carcinoma in situ is typically associated with excellent clinical outcomes and differs prognostically from invasive cancers. However, our decision to include individuals with stage 0 disease was based on the primary focus of this study, which centers on psychosocial and behavioral outcomes rather than clinical prognosis.
To clarify this rationale, we have added the following paragraph to the Discussion section (page 11, lines 335-344):
"Although carcinoma in situ (stage 0) is associated with an excellent prognosis, individuals with this diagnosis were included in the present study given the behavioral focus of our investigation. As formally recognized cancer survivors, patients with stage 0 disease may still experience substantial psychological distress—such as FCR—and encounter various unmet informational or emotional needs. Their inclusion reflects the full psychosocial spectrum of survivorship and supports a more comprehensive understanding of the factors that influence engagement in follow-up care across all cancer stages. Thus, their inclusion is justified in psychosocial and behavioral cancer research, as it ensures a more complete representation of the survivorship experience across the full spectrum of disease stages."
We hope this explanation adequately addresses the reviewer’s concern and supports the inclusion of these patients in our study sample.
- According to the clinical classification of tumors breast, lung, colorectal, etc. (as updated by the American Joint Committee on Cancer (AJCC) in 2018 for breast cancer staging), stages are represented by Roman numerals (I-IV), while the degree of histological malignancy (grade) is expressed using Arabic numerals (1-3). check the link:http://www.breastsurgeonsweb.com/wp-content/uploads/downloads/2020/10/AJCC-Breast-Cancer-Staging-System.pdf.
Response) Thank you for your careful review and valuable suggestion. As recommended, we have revised the cancer staging terminology throughout the manuscript to align with the AJCC (8th edition) guidelines. Specifically, Roman numerals (I–IV) are now used to represent clinical stage, whereas Arabic numerals (1–3) are used for histological grade where applicable.
The following revisions were made:
- page 4, line 127: "stage 3" was changed to "stage III."
- page 5, lines 185-186: Stage notations were updated to Roman numerals.
- In Table 1 (page 6), Table 2 (page 7), and Table 3 (page 8), the cancer stage entries have been revised accordingly.
We appreciate the reviewer’s attention to this important detail, which has helped improve the clarity and consistency of the manuscript.
- In the study group, the largest population consisted of women with breast cancer. I suggest adding information on the molecular (biological) subtype of breast cancer to Table.
Response) Thank you for your insightful suggestion. We agree that the biological subtypes of breast cancer may have important implications for psychological outcomes and follow-up behaviors. However, molecular subtype data (e.g., hormone receptor status and HER2 expression) were not consistently available for all breast cancer participants in our dataset. Given that the primary aim of this study was to examine behavioral rather than clinical subtype outcomes, we were unable to incorporate this information into the current analysis. We acknowledge this as a limitation and suggest that future research focusing on breast cancer survivors consider including molecular subtype data to enhance specificity and clinical relevance.
- Please add a paragraph in the discussion section explaining how the results can be applied in practice. Additionally, clarify why your work is valuable in the field and whether any general recommendations can be made based on your research.
Response) Thank you for your thoughtful comment. In response, we have added a paragraph to the Discussion section to clarify the practical implications of our findings and to emphasize the relevance of our work in the context of cancer survivorship care. The newly added paragraph also outlines general recommendations that may be applicable to broader survivor populations. Specifically, it highlights the potential use of unmet need assessments in clinical settings to identify individuals at risk of poor follow-up care adherence and suggests that comprehensive, psychosocially informed interventions could improve long-term health outcomes.
The following paragraph has been added to the Discussion section (page 11, lines 325-334):
“The findings of this study offer important implications for clinical practice and survivorship care. Identifying and addressing unmet needs—particularly those related to psychological distress and informational deficits—may enhance adherence to recommended screenings among survivors experiencing FCR. Incorporating routine assessments of unmet needs into survivorship care may help clinicians identify individuals at greater risk of disengagement from follow-up care. Furthermore, comprehensive, multidisciplinary interventions that target emotional and practical support needs could mitigate the psychological burden affecting health behaviors. These insights support a patient-centered, need-responsive model of follow-up care that may be applicable across diverse cancer survivor populations.”
We hope this addition adequately addresses the reviewer’s request and further enhances the practical value of the manuscript.
Round 2
Reviewer 2 Report
Comments and Suggestions for Authors
The authors have revised the article in accordance with the suggestions. I suggest publishing the article in its current form.